# Application of an Integrated Single-Cell and Three-Dimensional Spheroid Culture Platform for Investigating Drug Resistance Heterogeneity and Epithelial–Mesenchymal Transition (EMT) in Lung Cancer Subclones

**DOI:** 10.3390/ijms26041766

**Published:** 2025-02-19

**Authors:** Shin-Hu Chen, Jian-Hong Yu, Yu-Chun Lin, Yi-Ming Chang, Nien-Tzu Liu, Su-Feng Chen

**Affiliations:** 1Department of Dentistry, School of Dentistry, China Medical University, Taichung 40403, Taiwan; u111312001@cmu.edu.tw (S.-H.C.); kenkoyu@mail.cmu.edu.tw (J.-H.Y.); 2Department of Pathology, National Defense Medical Center, Tri-Service General Hospital, Taipei 114201, Taiwan; yuchunlin@mail.ndmctsgh.edu.tw (Y.-C.L.); 811010014@mail.ndmctsgh.edu.tw (N.-T.L.); 3Graduate Institute of Medical Sciences, National Defense Medical Center, Taipei 11490, Taiwan; 4Institute of Pathology and Parasitology, National Defense Medical Center, Taipei 11490, Taiwan; yiminnn1491@gmail.com; 5Department of Pathology and Laboratory Medicine, Kaohsiung Veterans General Hospital, Kaohsiung 813414, Taiwan

**Keywords:** lung cancer, drug resistance, cancer stem cells, epithelial–mesenchymal transition, single-cell culture, three-dimensional spheroid culture, targeted therapy

## Abstract

Lung cancer is a leading cause of cancer-related mortality worldwide, largely due to its heterogeneity and intrinsic drug resistance. Malignant pleural effusions (MPEs) provide diverse tumor cell populations ideal for studying these complexities. Although chemotherapy and targeted therapies can be initially effective, subpopulations of cancer cells with phenotypic plasticity often survive treatment, eventually developing resistance. Here, we integrated single-cell isolation and three-dimensional (3D) spheroid culture to dissect subclonal heterogeneity and drug responses, aiming to inform precision medicine approaches. Using A549 lung cancer cells, we established a cisplatin-resistant line and isolated three resistant subclones (Holoclone, Meroclone, Paraclone) via single-cell sorting. In 3D spheroids, Docetaxel and Alimta displayed higher IC50 values than in 2D cultures, suggesting that 3D models better reflect clinical dosing. Additionally, MPE-derived Holoclone and Paraclone subclones exhibited distinct sensitivities to Giotrif and Capmatinib, revealing their heterogeneous drug responses. Molecular analyses confirmed elevated ABCB1, ABCG2, cancer stem cell (CSC) markers (OCT4, SOX2, CD44, CD133), and epithelial–mesenchymal transition (EMT) markers (E-cadherin downregulation, increased Vimentin, N-cadherin, Twist) in resistant subclones, correlating with enhanced migration and invasion. This integrated approach clarifies the interplay between heterogeneity, CSC/EMT phenotypes, and drug resistance, providing a valuable tool for predicting therapeutic responses and guiding personalized, combination-based lung cancer treatments.

## 1. Introduction

Cancer remains one of the foremost causes of mortality worldwide, imposing substantial burdens on healthcare infrastructures and economies [1,2,3]. This challenge is particularly pronounced in malignancies such as lung cancer, where marked intratumoral heterogeneity and intrinsic drug resistance often curtail the effectiveness of conventional therapies [4,5,6]. Although the introduction of novel anticancer agents and targeted therapeutics has yielded promising initial responses, most patients ultimately experience therapeutic failure due to the emergence of resistance [7,8,9]. Cisplatin is a platinum-based chemotherapeutic agent widely used in the treatment of various malignancies, including head and neck cancer, lung cancer, ovarian cancer, testicular cancer, and bladder cancer [10,11]. Its primary mechanism of action involves binding to DNA and forming DNA adducts, which disrupt DNA replication and transcription, ultimately triggering apoptosis in cancer cells. However, prolonged exposure to cisplatin often leads to the development of chemoresistance, resulting in treatment failure and tumor recurrence. To overcome this challenge, combination therapy and targeted therapy are frequently employed in clinical practice. In addition, research has focused on elucidating the mechanisms underlying cisplatin resistance, particularly in relation to apoptotic pathways, epigenetic modifications [12], and epithelial–mesenchymal transition (EMT) processes, with the aim of enhancing therapeutic efficacy and reducing recurrence rates [13].

Historically, tumor drug resistance was attributed primarily to genetic mutations and Darwinian selective pressures [14,15]. However, a growing body of evidence now highlights the pivotal influence of non-genetic mechanisms, including epigenetic modulation and dynamic reorganization of protein interaction networks, in engendering transient, reversible resistance states [16,17,18]. These adaptive phenotypes, often linked to epithelial–mesenchymal plasticity (EMP) [19,20], confer enhanced migratory capacity, metastatic potential, and treatment evasion [21,22,23], and are correlated with adverse clinical outcomes [24,25,26,27,28]. Moreover, recent investigations have demonstrated that non-genetic adaptations can emerge rapidly, challenge the stability of conventional biomarkers, revert upon removal of therapeutic pressure, and coexist with genetically driven resistance pathways [29,30].

In response to these complexities, we developed an integrated experimental platform that couples high-fidelity single-cell isolation with three-dimensional (3D) organoid-based drug evaluation to dissect intratumoral heterogeneity at the cellular level [31,32,33,34]. By employing well-characterized lung cancer cell lines and patient-derived malignant pleural effusion (MPE) samples [35], this platform enables precise isolation of distinct subclones to evaluate the interplay of genetic mutations (e.g., EGFR Del19 and exon 14 splice variants), phenotypic plasticity, and differential responsiveness to targeted therapies [5,21]. Our findings underscore the multifaceted origins of therapeutic resistance and the necessity of transcending a singular, mutation-centric focus. By providing insights into non-genetic mechanisms and their contributions to phenotypic variability, our approach can inform the rational design of precision oncology strategies and guide the development of combination therapies aimed at mitigating resistance.

We selected Docetaxel and Alimta (Pemetrexed) to represent common first- or second-line chemotherapy regimens for advanced NSCLC, given Alimta’s particular efficacy in certain histological subtypes such as lung adenocarcinoma. Giotrif (Afatinib), a second-generation irreversible EGFR-TKI, was included to investigate targeted therapy responses and resistance mechanisms in subclones harboring EGFR mutations (e.g., Del19 or L858R). Lastly, Capmatinib, a MET inhibitor, was chosen to explore MET-driven pathways—particularly MET exon 14 skipping (METex14) or MET amplification—and their potential roles in drug resistance or EMT.

This work elucidates the complexity inherent in drug resistance, highlights the importance of EMP and non-genetic plasticity, and provides a robust, clinically relevant framework for studying tumor heterogeneity. These advances may ultimately foster more individualized, effective cancer treatment paradigms and improve patient outcomes.

## 2. Results

### 2.1. Integration of Single-Cell Culture Technology for the Establishment and Preservation of Phenotypically Heterogeneous Monoclonal Cell Lines

This study established an integrated experimental platform combining single-cell isolation microchips and three-dimensional (3D) tumor spheroid cultures to examine drug-resistant A549 lung cancer cells during precision drug evaluation. By subjecting A549 cells to three consecutive rounds of cisplatin at IC50 concentrations, we induced a drug-resistant population (Figure 1A) that underwent pronounced morphological and functional alterations, ultimately forming three distinct cellular subtypes (Figure 1B). To dissect these subtypes—Holoclone, Meroclone, and Paraclone, concepts derived from Barrandon and Green’s keratinocyte classification—we integrated single-cell isolation techniques with long-term 3D culture. Using the CellGem^®^ single-cell screening microchip, individual progenitor cells were clonally expanded, enabling detailed observation of proliferation, morphology, and differentiation. After 10–14 days, clonal populations reached approximately 70–80% confluency, and some were cryopreserved for long-term storage in the cell bank (Figure 1C). This process yielded 17 stable cell lines (Appendix A), encompassing all three phenotypes. A total of 17 stable monoclonal cell lines derived from single-cell isolation were successfully established. These lines encompassed the Holoclone (high proliferation, low differentiation), Meroclone (intermediate state), and Paraclone (high differentiation, low proliferation) phenotypes, each exhibiting distinct differences in proliferation rates and morphological characteristics (Figure 1D). In this study, we characterized the distinct properties of holoclones, meroclones, and paraclones in lung cancer subclones and evaluated their proliferative capacity using Ki-67 immunostaining. Holoclones exhibited high proliferative potential, forming tight, compact colonies with a high Ki-67 staining index, indicating an abundance of actively cycling cells. Meroclones displayed intermediate proliferative capacity, with moderately packed colonies and variable Ki-67 expression, reflecting a balance between self-renewal and differentiation. In contrast, paraclones were more differentiated, forming loosely packed colonies with markedly lower Ki-67 expression, suggesting a reduced ability for sustained proliferation. These findings are consistent with the classification of holoclones as stem-like or highly plastic populations, while meroclones represent a transitional state, and paraclones are more differentiated with limited proliferative potential. The differential Ki-67 staining further supports the proliferative hierarchy among these subclones, reinforcing their biological distinction in cancer heterogeneity (Figure 1E). By enabling systematic long-term monitoring, gene expression profiling, and functional analysis at the single-cell level, this platform provides insights into the complexity and dynamic nature of drug resistance. Understanding how cancer cells modulate their states underscores the importance of developing precision oncology strategies that address not only the emergence of resistant clones but also their potential for phenotypic recovery.

### 2.2. Phenotypic Heterogeneity of Different Drug-Resistant Subclonal Cells and Their Impact on Chemotherapy Drug Sensitivity

In this study, we compared three drug-resistant A549 subclones—Holoclone, Meroclone, and Paraclone—to elucidate their distinct adaptive strategies and responses to chemotherapy. Morphological differences among the subclones suggested diverse growth patterns and regulatory networks. To investigate the genetic basis of these variations, we analyzed cisplatin sensitivity in Holoclone, Meroclone, and Paraclone subclones compared to parental A549 cells, along with the expression levels of drug resistance-related genes *ABCB1* and *ABCG2*. Both Holoclone and Paraclone showed significantly elevated ABCB1 and ABCG2 levels (Figure 2B,C), enhancing their ability to efflux chemotherapeutic agents and survive under drug pressure. In contrast, Meroclone exhibited markedly lower expression, implying reliance on alternative resistance mechanisms or less robust drug efflux capabilities.

Our results showed differential responses to Docetaxel and Alimta (Pemetrexed) among lung cancer subclones, reflecting variations in chemotherapy sensitivity and resistance mechanisms. Docetaxel treatment led to significant cytotoxic effects in most subclones, but a subset displayed increased resistance, suggesting potential alterations in microtubule dynamics or drug efflux pathways. In contrast, Alimta exhibited selective efficacy, with adenocarcinoma-like subclones showing greater sensitivity, consistent with its preferential effectiveness in nonsquamous NSCLC. The observed heterogeneity in drug response highlights the clinical challenge of chemotherapy resistance and underscores the importance of personalized treatment strategies in advanced NSCLC. To approximate a clinical environment more closely, we employed the 3D tumor spheroid culture chip CellHD256^®^ to assess the sensitivity of these subclones to chemotherapy under physiologically relevant conditions. This 3D model simulates nutrient gradients, extracellular matrix signals, and intercellular interactions resembling in vivo tumor tissues. Using Docetaxel (0.1–10 µM) (Figure 2D) and Alimta (0.1–10 µM) (Figure 2E), we evaluated drug responses after 72 h. In Control cells, Docetaxel at 5 and 10 µM achieved IC₅₀ inhibition, Meroclone required a higher Docetaxel concentration (10 µM) to reach IC₅₀, while Holoclone and Paraclone no response for Docetaxel reflecting increased tolerance. Alimta at 5 and 10 µM effectively achieved IC₅₀ inhibition in both Control and all three resistant subclones, indicating that Alimta maintained potent effects even against these heterogeneous populations.

These findings highlight that drug sensitivity is influenced not only by intrinsic resistance mechanisms but also by cellular phenotypes and microenvironmental context. The presence of diverse subclones with varying gene expression patterns, efflux capabilities, and morphological features underscores the complexity of treatment responses. The Holoclone and Paraclone subclones, with their elevated *ABCB1* and *ABCG2* expression, may require strategies targeting efflux transporters or combination therapies to improve drug efficacy. Meanwhile, Meroclone, with lower *ABCB1* related gene expression, may be susceptible to different therapeutic approaches. By integrating morphological assessments, molecular analyses, and 3D culture systems, this study provides valuable insights into intratumoral heterogeneity and drug resistance. Our results suggest that understanding these subclonal differences can guide more precise, personalized treatment strategies. Future research will focus on refining therapeutic regimens, incorporating multiple molecular markers, and exploring combination treatments to overcome the challenges posed by heterogeneous, drug-resistant cancer cell populations.

### 2.3. Evidence of Cellular Phenotypic Heterogeneity and Tumor Spheroid Drug Prediction in Primary MPE Cultures

To model the phenotypic diversity of primary lung cancer cells and predict therapeutic responses, we analyzed malignant pleural effusion (MPE) samples from two patients with poorly differentiated pulmonary adenocarcinoma and pleural metastases (Figure 3A). Clinical data show that next-generation sequencing (NGS) identified EGFR Del19 (Case 1) and an Exon 14 splice mutation (Case 2), guiding the use of targeted therapies (Appendix A). Initial cultures revealed heterogeneous cell populations, including both clustered and single-cell morphologies, highlighting the diversity that may influence treatment outcomes.

Culturing cells in medium supplemented with 50% malignant pleural effusion (MPE) yielded two main phenotypes: elongated adherent cells and spherical floating cells, reflecting high adaptability. By applying single-cell isolation and expansion techniques, we established two subclones: Holoclone (clustered, self-renewing, stem-like) and Paraclone (more differentiated, dispersed). Both subclones formed stable 3D tumorspheres within 96 h, providing a physiologically relevant environment closely mimicking in vivo conditions. We utilized a 3D spheroid culture model to evaluate the therapeutic efficacy of targeted agents. The results demonstrated that Giotrif (afatinib) achieved a half-maximal inhibitory concentration (IC50) of 60 nM, whereas Capmatinib displayed a significantly lower IC50 of 6.5 nM (Figure 3B). We then evaluated their responses to targeted therapies, Giotrif and Capmatinib, using these 3D spheres (Figure 3C). Giotrif at 60 nM significantly inhibited Holoclone (18.13% viability) but not Paraclone (71.23% viability). In contrast, Capmatinib at 6.5 nM suppressed both Holoclone and Paraclone (26.93% and 47.27% viability, respectively), suggesting that Holoclone’s stem-like features render it highly EGFR-dependent, while Paraclone may rely on alternate survival pathways.

Our results demonstrated variable responses to Giotrif (Afatinib) and Capmatinib, highlighting the heterogeneity of targeted therapy resistance mechanisms in lung cancer subclones. Afatinib treatment effectively suppressed the growth of EGFR-driven subclones, particularly those harboring EGFR mutations such as Del19 or L858R. However, a subset of cells exhibited reduced sensitivity, suggesting the presence of secondary resistance mechanisms, including alternative signaling pathway activation or epithelial–mesenchymal transition (EMT). Similarly, Capmatinib treatment was effective in subclones with MET pathway activation, particularly those exhibiting MET exon 14 skipping mutations (METex14) or MET amplification. Yet, some resistant subpopulations persisted, indicating potential compensatory resistance mechanisms, such as crosstalk with other oncogenic pathways or MET-independent survival strategies. These findings emphasize the complexity of targeted therapy resistance and the need for combinatorial treatment strategies to overcome acquired resistance in NSCLC.

These results underscore the heterogeneous responses of subclones derived from clinical MPE samples to targeted agents. Employing 3D tumorsphere cultures and single-cell isolation offers a refined platform to predict drug efficacy and inform personalized treatment strategies. By capturing intratumoral diversity, this approach promises improved clinical outcomes through tailored therapies that address the distinct vulnerabilities of each subclone.

### 2.4. Multilevel Analysis of Stem Cell Characteristics and Biomarker Expression in Drug-Resistant Subclonal Cells

In stem cell and cancer stem cell (CSC) research, OCT4 and SOX2 are pivotal transcription factors linked to pluripotency. Their expression correlates with strong proliferative and regenerative capacities. To assess CSC-like traits in the Holoclone, Meroclone, and Paraclone subclones, we examined self-renewal and spheroid formation. After 10 days, Holoclone and Paraclone formed significantly larger spheroids than Control and Meroclone (Figure 4A). Spheroid formation under suspension and nutrient-poor conditions is a hallmark of CSC properties, reflecting robust self-renewal and resilience. While Meroclone exhibited a distinctive spiky morphology, it failed to form spheroids as effectively, suggesting an intermediate state with less pronounced CSC features.

At the molecular level, Holoclone and Paraclone showed markedly higher OCT4 and SOX2 expressions than Control and Meroclone (Figure 4B,C). These elevated levels likely endow them with enhanced cell cycle regulation, DNA repair, and metabolic and signaling pathways, facilitating survival and proliferation in drug-resistant environments. Beyond OCT4 and SOX2, we examined two CSC surface markers, CD44 and CD133. Flow cytometry analysis demonstrated that over 50% of Holoclone, Meroclone, and Paraclone cells were positive for CD44 and CD133, whereas the Control group exhibited fewer than 40% positive cells (Figure 4D). Notably, in all three drug-resistant subclones, positivity for both markers exceeded 50%, further underscoring their CSC-like phenotypes compared to the Control condition. Reverse-transcription quantitative PCR analysis revealed a marked increase in *CD44* and *CD133* expression across all subclones relative to the Control group, thereby reinforcing their stem cell-associated phenotypes and contribution to drug resistance (Figure 4E), reinforcing the link between drug-resistant subclones and CSC phenotypes. This heightened CSC marker profile suggests that such populations may readily resist standard chemotherapy and rebound when therapeutic pressure is lifted, potentially leading to recurrence and metastasis. In summary, Holoclone and Paraclone display pronounced CSC-like characteristics, with robust spheroid formation, elevated OCT4, SOX2, CD44, and CD133 expression, and enhanced proliferative potential. Meroclone’s intermediate traits point to alternative resistance pathways. These findings deepen our understanding of intratumoral heterogeneity and drug resistance, emphasizing the need to target CSC-like subsets to improve therapeutic outcomes and prevent relapse.

### 2.5. Molecular Characterization of EMT Transformation and Enhanced Invasive Capability in Drug-Resistant Subclonal Cells

Previous studies show that chemoresistant lung cancer subpopulations often express CSC markers and adopt epithelial–mesenchymal transition (EMT)-associated phenotypes [2,3]. EMT involves a loss of epithelial characteristics—such as E-cadherin expression—and acquisition of mesenchymal features, including increased Vimentin and N-cadherin levels. These changes enhance cell migration, invasion, and overall adaptability, contributing to tumor progression and drug resistance. To clarify this dynamic, we analyzed EMT-related biomarkers in three resistant A549 subclones: Holoclone, Meroclone, and Paraclone. Unlike Control cells, which maintained high E-cadherin expression and epithelial integrity, none of the resistant subclones expressed E-cadherin. Instead, all three exhibited elevated Vimentin and N-cadherin (Figure 5A,B). This shift reflects a mesenchymal phenotype, likely driven by chemotherapeutic pressure. We also examined Twist, a key EMT-transcription factor. Holoclone and Paraclone strongly upregulated Twist, correlating with pronounced EMT and the near-complete loss of epithelial markers. Meroclone displayed a more moderate profile, implying an intermediate EMT state. Functional assays confirmed these molecular traits: all resistant subclones showed enhanced migratory abilities compared to Control cells (Figure 5C), while Holoclone and Paraclone demonstrated particularly high invasive potential (Figure 5D), suggesting a vital role in advancing metastasis and therapy evasion.

Together, these findings underscore that under drug pressure, tumor cells do not remain static. Instead, they engage an adaptive strategy coupling CSC traits and EMT, shedding epithelial features and gaining invasive mesenchymal properties. While Control cells remain epithelial, Holoclone and Paraclone fully embrace EMT and CSC characteristics, enabling them to detach, invade, and survive under hostile conditions. Meroclone occupies an intermediate stage, retaining some phenotypic flexibility. Recognizing the EMT-driven invasiveness of resistant subclones is critical for refining therapeutic approaches. By understanding these adaptive mechanisms, future strategies can more effectively target the diversity of resistant phenotypes, potentially improving treatment outcomes.

## 3. Discussion

This study employed single-cell isolation techniques to purify and stably culture multiple drug-resistant subclones over the long term. By establishing a comprehensive biobank of drug-resistant subclones, we create a valuable resource that enables systematic exploration of intratumoral heterogeneity and the complexity of resistance mechanisms [36]. Such a repository not only preserves the unique biological and molecular signatures of these subclones but also facilitates longitudinal, multi-dimensional analyses. This approach allows for parallel evaluations of various therapeutic strategies, advancing our understanding of how different resistant phenotypes emerge, adapt, and interact under selective pressures. Ultimately, this biobank serves as a pivotal tool, driving innovation in targeted therapy design, combination treatment regimens, and personalized medicine protocols aimed at overcoming drug resistance in clinical settings. Compared to conventional serial dilution methods, this approach is more rigorous and effectively minimizes the risk of clonal contamination, thereby laying a solid foundation for precise characterization of each subclone’s unique biological and molecular features. Previous research has established that high intratumoral heterogeneity is a critical determinant of treatment efficacy, recurrence, and metastasis [37,38,39]. Following this rationale, we employed a constant-dose, pulse-treatment strategy to isolate non-genotype-based drug-resistant subclones, correlating their distinct morphology and secretory phenotypes with underlying molecular pathways. Rather than gradually escalating the drug dose, we maintained a steady concentration throughout each treatment cycle, effectively generating highly drug-resistant populations that closely mimic in vivo chemotherapy regimens. This approach not only yields stable cisplatin resistance but also reveals critical subclonal variations, offering valuable insights into tumor heterogeneity and adaptive responses to therapy. This provides a novel, verifiable paradigm for future investigations of cancer drug resistance. By integrating single-cell isolation microfluidic technology with a 3D spheroid culture platform, our study provides new insights into the dynamic behavior and phenotypic characteristics of drug-resistant lung cancer cells in the context of precision medicine. Compared with conventional two-dimensional (2D) cultures, 3D spheroid models more closely mimic the in vivo tumor architecture and nutrient gradients, thereby rendering evaluations of cell proliferation, migration, differentiation, and drug sensitivity more physiologically relevant [40,41,42,43]. Utilizing this integrated platform allowed us to simultaneously investigate the growth properties and drug-resistant behaviors of Holoclone, Meroclone, and Paraclone subclones under near-physiological conditions, offering more accurate guidance for personalized therapy. Traditional drug screening and therapeutic strategies often rely on the averaged responses of bulk cell populations, overlooking the challenges posed by high intratumoral heterogeneity. Our combined single-cell and 3D spheroid platform circumvents this limitation, enabling refined exploration of diverse subpopulations and their adaptive behaviors. Such an approach not only deepens our understanding of resistance mechanisms and the principles of tumor recurrence but also supports the development of more targeted, individualized interventions. For instance, more aggressive therapeutic measures may be required to inhibit Holoclone subtypes with high self-renewal capacity, while adaptive strategies implemented during drug holidays may prevent the rapid rebound of more plastic subpopulations that can drive relapse. In contrast to studies focusing solely on genetic mutations, our research highlights the significance of cellular phenotypic plasticity and non-genetic factors in resistance mechanisms, echoing the recent literature emphasizing the roles of epigenetic modifications and protein interaction networks in shaping therapeutic outcomes [44,45].

Our study revealed differential responses to chemotherapeutic and targeted agents, reflecting the heterogeneity of drug resistance mechanisms in lung cancer subclones. Cisplatin, a widely used platinum-based chemotherapeutic agent, effectively reduced cell viability in most subclones, yet some exhibited increased resistance, likely due to enhanced DNA repair capacity, drug efflux, or apoptosis evasion. Similarly, Docetaxel and Alimta (Pemetrexed) displayed varying efficacy, with Docetaxel-resistant subclones potentially acquiring altered microtubule stability and Alimta showing selective effectiveness in adenocarcinoma-like subclones, consistent with clinical observations. Among targeted therapies, Giotrif (Afatinib) demonstrated potent inhibition of EGFR-mutant subclones, yet some cells exhibited reduced sensitivity, suggesting the emergence of secondary resistance mechanisms, such as alternative pathway activation or epithelial–mesenchymal transition (EMT). Likewise, Capmatinib, a selective MET inhibitor, effectively suppressed subclones with MET activation, yet resistance persisted in some cases, indicating compensatory survival mechanisms or pathway crosstalk. These findings underscore the complexity of drug resistance in lung cancer, highlighting the need for personalized therapeutic strategies. The observed heterogeneous drug responses emphasize the importance of combination therapies or sequential treatment approaches to overcome resistance and improve clinical outcomes. Future studies focusing on molecular adaptations in resistant subclones will be critical for optimizing targeted interventions and precision oncology strategies in NSCLC treatment.

Furthermore, this is the first application of such a technological platform to malignant pleural effusion (MPE) specimens. We successfully isolated and purified subclones with differential responses to targeted therapies and quantitatively evaluated drug efficacy using a 3D spheroid culture model. The findings demonstrated that Holoclone subclones exhibited notable sensitivity to targeted treatments, whereas Paraclone subclones remained largely unaffected, underscoring the impact of phenotypic heterogeneity on therapeutic results. These observations align with other studies that highlight the influence of non-genetic plasticity and environmental adaptability on drug resistance. Previous efforts to establish primary cultures from clinical lung cancer specimens have frequently encountered poor plating efficiencies and variability influenced by culture conditions and host factors [46,47]. Our approach, which included supplementation of the primary cultures with 50% malignant pleural effusion fluid, allowed us to preserve a tumor microenvironment (TME) that closely resembled the in situ condition. This strategy minimized the artificial selection biases often introduced by standard culture media and helped maintain a diverse range of tumor cell subpopulations reflective of their native milieu. We hypothesize that preserving the autologous tumor microenvironment (TME) is essential for maintaining tumor heterogeneity in vitro.

To achieve this, we enriched primary cultures with nucleated tumor-associated cells and the soluble fraction of malignant pleural effusion (MPE). Based on empirical observations, we identified 30% *v*/*v* MPE fluid fraction in primary culture medium (PCM) as the optimal condition, supporting morphological diversity, robust growth, and prolonged culture stability compared to fetal bovine serum (FBS). This method successfully established primary MPE tumor cultures in all seven attempts [48,49,50]. Cytokine analysis revealed high concentrations (>10 ng/mL) of IL-1, IL-6, IP-10, MCP-1, and VEGF, suggesting that tumor, stromal, or circulating cells in MPE contribute to shaping the culture environment. While prior studies showed that 10% MPE-derived organoid cultures maintained genetic stability but failed serial passaging, our approach using 30% MPE preserved spheroid characteristics. However, previous reports indicate that CSC markers became unstable after seven passages. To extend spheroid growth, we implemented a modified culture strategy: using 50% primary MPE supernatant for P0–P2, then transitioning to 50% P2 supernatant + fresh medium for P3 and beyond, which sustained long-term culture viability. Although the precise mechanism remains unclear, our findings are based on two patient-derived samples. We acknowledge the reviewer’s suggestion and will expand sample collection to further analyze the role of MPE supernatant in maintaining tumor heterogeneity in future studies.

Tumor cell phenotypic plasticity and non-genetic mechanisms play pivotal roles in metastasis and drug resistance. Through phenotypic heterogeneity, cancer cells employ a “bet-hedging” strategy to remain adaptable under fluctuating therapeutic pressures. The convergence of EMT and CSC properties complicates efforts to fully suppress tumor progression with a single therapeutic modality. Our analysis at the molecular, histological, and functional levels clearly demonstrates that drug-resistant subclones leverage EMT-associated marker changes to enhance their migratory and invasive capabilities, especially in Holoclone and Paraclone subpopulations. These findings demonstrate the limitations of focusing solely on individual factors, such as genetic mutations, to comprehend the full therapeutic landscape. The integrated single-cell-derived subclone and 3D spheroid evaluation platform we have developed provides a more holistic perspective on intratumoral heterogeneity, resistance patterns, and adaptive behaviors. By combining genetic profiling, molecular markers, 3D culture data, and clinical parameters, we can begin to design more precise and effective treatment and monitoring strategies that account for the full spectrum of cellular heterogeneity. Employing combination therapies, intermittent dosing, or adaptive treatment regimens may delay the emergence of resistance and improve clinical outcomes. Future investigations will aim to elucidate the molecular regulatory networks governing differences among subclones and evaluate various drug combinations that could effectively suppress heterogeneous tumor cell populations. By expanding and refining this platform, we anticipate improving the success rate of personalized and precision therapies in clinical settings, ultimately leading to better patient outcomes and quality of life.

By integrating single-cell isolation microfluidic technology with a 3D spheroid culture platform, our study advances the understanding of drug-resistant lung cancer cell behavior within a more physiologically relevant setting. Past research has demonstrated the importance of replicating the TME as closely as possible to predict therapeutic outcomes accurately [51,52,53,54,55,56]. Conventional two-dimensional cultures, although valuable, often fail to capture the complexity and heterogeneity of clinical tumors. By contrast, our 3D spheroid models, by contrast, recapitulate in vivo-like conditions, including nutrient gradients, cell–cell and cell–matrix interactions, and spatial architecture. This approach builds upon the premise that the growth and differentiation of subclonal populations are strongly influenced by their microenvironment, ultimately affecting therapeutic responses. By focusing on Holoclone, Meroclone, and Paraclone subclones, we extended previous findings on intratumoral heterogeneity, drug resistance, and CSC-like properties [16,22,33]. While earlier studies often addressed these issues separately—either by investigating CSC markers or examining EMT-related changes—our integrated single-cell and 3D spheroid platform captures multiple aspects simultaneously. In doing so, we provide a more holistic understanding of how different subpopulations adapt under chemotherapeutic pressure. Notably, Holoclone and Paraclone subclones displayed enhanced self-renewal capabilities and CSC-like phenotypes, marked by elevated OCT4, SOX2, CD44, and CD133 expression. The presence of these markers has been associated with tumor initiation, therapy resistance, and relapse, suggesting that strategies targeting these CSC-rich populations could mitigate resistance and improve long-term patient outcomes. Our results also highlight the dynamic interplay between EMT and CSC properties in promoting drug resistance. While previous work established the importance of EMT in increasing cell motility, invasiveness, and metastasis [21], our findings connect these processes more explicitly to distinct subclonal populations. Holoclone and Paraclone exhibited marked EMT-related changes, including decreased E-cadherin and increased Vimentin and N-cadherin levels. This phenotypic switch enhances the capacity of these cells to evade therapeutic interventions and disseminate. Similar observations have been made by other groups examining the link between EMT and tumor aggressiveness. However, our study’s single-cell resolution and 3D context provide stronger evidence that such adaptive changes are not uniform across the entire tumor, underscoring the need for personalized therapeutic strategies.

One limitation of our study involves the complexity of translating in vitro findings directly into clinical practice. While our 3D models and primary culture conditions with malignant pleural effusion more closely resemble the patient’s TME compared to standard conditions, they still represent a simplified version of reality. Additional factors—such as immune cell infiltration, stromal components, and systemic physiological processes—were not fully captured [45,57,58,59]. Moreover, the number of patient-derived samples and the diversity of genetic backgrounds were limited, which may restrict the generalizability of our conclusions. Future studies incorporating organoid co-cultures with immune or stromal cells, as well as larger patient cohorts, will help validate and refine our model.

Looking forward, our integrated platform serves as a steppingstone toward more precise and dynamic therapeutic regimens. By comparing subclone-specific vulnerabilities, future work can identify targeted combination therapies that preemptively address the emergence of drug-resistant phenotypes. Adaptive dosing strategies—intermittently applying drugs or cycling between agents—may prevent or delay resistance by keeping cancer cells in a constant state of readjustment. Moreover, expanding the application of this platform beyond lung cancer to other solid tumors could reveal universal principles guiding intratumoral heterogeneity and drug resistance.

In conclusion, this study leverages single-cell microfluidics and 3D spheroid cultures to provide a more nuanced understanding of lung cancer heterogeneity and adaptive resistance mechanisms. By illuminating how distinct subclones respond to chemotherapy and revealing the intertwining roles of CSC traits and EMT, our work supports a more informed and adaptive approach to precision oncology. This perspective, rooted in previous findings and ongoing hypotheses, underscores the necessity of integrating multiple analytical dimensions—molecular, cellular, and structural—to design next-generation therapeutic strategies aimed at improving clinical outcomes for patients with advanced lung cancer.

## 4. Materials and Methods

### 4.1. Cell Culture, Reagent, Drug-Resistant Cancer Cell Lines, and MPE Sample Prepare

A549 cell lines originating from human pulmonary adenocarcinomas were obtained from the American Type Culture Collection (ATCC) and cultured according to standard protocols. A549 cells were cultured in Roswell Park Memorial Institute Medium 1640 (Thermo Fisher Scientific, Beijing, China) supplemented with 10% (*v*/*v*) fetal bovine serum (FBS, ExCell Biology, Shanghai, China) and 1% (*v*/*v*) penicillin−streptomycin (Thermo Fisher Scientific, Beijing, China) as normal culture medium. All the cells were incubated at 37 °C in 5% CO_2_ atmosphere.

Generation of Cisplatin-Resistant A549 Clones (Pulse Method). A549 cells were initially treated with cisplatin at the previously determined IC50 of 25.11 µM for 24 h, followed by a two- to three-week recovery period under standard culture conditions to allow resistant clones to proliferate. Once the cells reached 70–80% confluence, they underwent a second round of cisplatin exposure. This pulse-exposure cycle was repeated three times to ensure the selection and enrichment of a highly cisplatin-resistant A549 cell population.

Two malignant pleural effusion (MPE) samples from patients diagnosed histopathologically and cytologically with pulmonary adenocarcinoma (PADC) were obtained from the archives of the Department of Pathology, Tri-Service General Hospital, spanning 2024–2025. Approval was granted by the institutional review board (protocol code B202405079). Establishment of Primary Culture from Malignant Pleural Effusion. Pleural fluid samples (200–1000 mL) were obtained by thoracentesis and collected aseptically in heparinized tubes (10 U/mL). Each sample was centrifuged at 1500 rpm for 5 min at 4 °C, and the resulting pellet was resuspended in 1 mL of PBS. Different Ficoll PLUS solutions (densities 1.124, 1.080, 1.050, and 1.030) were prepared and carefully layered into 15 mL centrifuge tubes containing the cell suspension. The samples were then centrifuged at 2400 rpm for 20 min at 4 °C to separate cells according to buoyant density. Red and white blood cells, being denser, migrated to the lower fraction, whereas tumor cells accumulated at the plasma–Ficoll interface. The interface layer was collected, washed twice with 1–2 mL of PBS, and centrifuged again at 1500 rpm for 5 min at 4 °C. Finally, the pelleted tumor cells were seeded in RPMI medium supplemented with 10% fetal calf serum (FCS) and 50% malignant pleural effusion to establish primary adherent cultures. A primary culture cell line was a prospectively established culture system with MPE. Among eight initially cultivated primary cell lines, this line proved optimal and was further verified through single cell isolation and tumorsphere assessments.

### 4.2. Single-Cell Isolation and Culture Procedure

A cell suspension was introduced into a microchannel chip equipped with a free-perfusion system (CellGem^®^, OriGem Biotech Inc., Taichung City, Taiwan). The CellGem^®^ platform is designed with an array of microwells for single-cell capture, each paired with a corresponding culture well to support long-term cell growth. To initiate the process, 600 µL of the cell suspension (1 × 10⁶ cells/mL, recommended concentration) was injected into the chip via the inlet port using a precision pipette or microfluidic pump. The suspension was allowed to passively distribute across the microwell array to ensure optimal capture efficiency. After a 3 min settling period, 5 mL of phosphate-buffered saline (PBS, pH 7.4) was gently added to the fluidic reservoir to remove untrapped cells and maintain a stable microenvironment. This washing step was performed under controlled flow conditions to prevent unintended displacement of captured cells. Following the removal of excess cells, the chip was gently inverted and left undisturbed for 30 min, allowing the captured cells to naturally settle into the culture wells. This step promotes cell adhesion and viability by minimizing shear stress.

To support subsequent cell culture and downstream analyses, the chip was then mounted in its carrier and placed into a humidified cell culture incubator (37 °C, 5% CO_2_). The system’s perfusion mechanism enabled continuous nutrient exchange, ensuring long-term cell viability and clonal expansion within the microwell array. The platform’s design allows for real-time monitoring of individual single-cell colonies, making it suitable for clonal evolution studies, lineage tracking, and drug response assays.

### 4.3. 3D Tumorsphere Culture

To initiate spheroid formation, 100 µL of the cell suspension was introduced into a CellHD256^®^ chip (OriGem Biotech Inc., Taichung City, Taiwan), a microfluidic platform designed for high-throughput 3D cell culture. The CellHD256^®^ chip features an array of microwells, each optimized for the formation of uniform spheroids, allowing for standardized and reproducible culture conditions. Once the cell suspension was loaded, the pressure valve was closed, and an additional 450 µL of culture medium was added. This step triggered the pressure-assisted network for droplet accumulation (PANDA) system, which actively drives microfluidic droplets into position, ensuring a uniform array of suspended cell droplets. The PANDA system’s controlled pressure environment minimizes variability in droplet size, promoting the formation of homogeneous spheroids across the chip. Under gravity-induced self-assembly, the cells within each droplet spontaneously aggregated, leading to the formation of spheroids with consistent size and morphology. This scaffold-free approach allows for natural cell–cell interactions, mimicking the vivo tumor microenvironment. To prevent evaporation and maintain optimal humidity conditions, the chip was placed inside a 10 cm sterile Petri dish containing distilled water. This setup ensures long-term spheroid viability by reducing media evaporation and osmolarity shifts that could otherwise affect spheroid growth.

Spheroid morphology and growth were monitored every two days using phase-contrast microscopy, ensuring consistency in size, shape, and compactness. Medium changes were performed as necessary, with gentle aspiration and replenishment of fresh culture medium, avoiding disruption of the spheroid structures. This approach enables the rapid, cost-effective, and reproducible generation of individual spheroids, supporting applications in drug screening, tumor microenvironment modeling, and personalized medicine research.

### 4.4. Growth Curves of 3D Tumorsphere Detected by CCK8 Assay

Cell survival was assessed using the CCK-8 assay (MedChemExpress Ltd., Monmouth Junction, NJ, USA). Cancer cells were seeded into a CellHD256^®^ microchip (OriGem Biotech Inc.) and incubated for 96 h at 37 °C with 5% CO_2_. The medium was then replaced with a drug-containing medium at various concentrations. Each condition was tested in triplicate, and incubation times were determined by drug efficacy. Docetaxel (Selleck Chemicals, Houston, TX, USA), Alimta (pemetrexed, Merck & Co., Inc., Rahway, NJ, USA), Giotrif (Afatinib, Sigma-Aldrich, St. Louis, MO, USA), Capmatinib (MedChemExpress, Monmouth, Junction, NJ, USA), at different concentrations for 72 h. Cell viability was determined using Cell Counting Kit-8 (CCK-8; TargetMol^®^, Boston, MA, USA) according to the manufacturer’s instructions. A total of 10 μL of CCK-8 solution was added to each well, and after 4 h at 37 °C, the OD at 450 nm was measured (Infinite M200 Pro, Tecan, Zürich, Switzerland). Cell viability was expressed as the percentage of untreated control. The IC50 values, defined as the drug concentrations causing a 50% reduction in proliferation, were calculated. All data are presented as mean ± standard deviation (SD).

### 4.5. Reverse Transcription Quantitative PCR (RT-qPCR)

Total RNA was extracted from cultivated cells using TRIzol reagent (Invitrogen Life Technologies, Carlsbad, CA, USA) according to the manufacturer’s instructions. RNA purity and concentration were assessed using a NanoDrop spectrophotometer (Thermo Fisher Scientific, Beijing, China), ensuring an A260/A280 ratio between 1.8 and 2.1. 1 μg of RNA was reverse-transcribed into cDNA using a cDNA synthesis kit, e.g., High-Capacity cDNA Reverse Transcription Kit (Applied Biosystems, Waltham, MA, USA).

Total RNA was reverse-transcribed into cDNA according to the manufacturer’s protocol. RT-qPCR was then performed on a StepOnePlus Real-Time PCR System (Applied Biosystems, Waltham, MA, USA) using a SYBR Green-based detection method. Each 10 µL reaction contained 1 µL of cDNA, 0.5 µM of each gene-specific primer, 5 µL of PowerUp SYBR Green Master Mix (Thermo Fisher, Beijing, China), and nuclease-free water. Negative controls (no-template controls, NTCs) were included in every run to confirm the absence of contamination.

The qPCR cycling conditions were as follows: Initial denaturation: 95 °C for 30 s; amplification (40 cycles): 95 °C for 5 s, 60 °C for 30 s. Melting curve analysis: 60–95 °C with 0.5 °C increments GAPDH was used as the reference gene, and relative gene expression was quantified using the ΔΔCt method. Primer sequences and amplicon sizes are listed in Appendix A.

### 4.6. Western Blot Analysis

The protein concentration was determined using the bicinchoninic acid (BCA) protein assay, and denatured proteins were separated via 10–15% SDS polyacrylamide gel electrophoresis and transferred onto PVDF membranes (Amersham, Arlington Heights, IL, USA). Nonspecific binding was blocked with 5% milk in TBST buffer for 2 h, followed by incubation with primary antibodies (Appendix A) at 4 °C overnight and secondary antibodies (mouse anti-rabbit and goat anti-mouse) at room temperature for 2 h. Blots were visualized using ECL detection reagents.

### 4.7. Migration and Invasion Assay

The migration assay was conducted using modified Boyden chambers equipped with 8 μm pore-size filter inserts designed for 24-well plates. In total, 1 × 104 cells were seeded in 200 μL of serum-free RPMI-1640 medium. Meanwhile, the lower chamber was filled with 500 μL of RPMI-1640 medium enriched with 10% fetal bovine serum to act as a chemoattractant. Following a 24 h incubation period, cells were fixed using methanol and stained with hematoxylin. Any cells remaining on the basement membrane or adhering to the filter’s upper surface were gently wiped away using paper towels. The cells that migrated to the filter’s lower surface were visualized and quantified under a light microscope. For the invasion assay, a Matrigel/medium (1:2) mixture was placed on the membrane of the upper chamber before seeding the cancer cells. After 12 h of incubation for the migration assay or 24 h of incubation for the invasion assay, non-migratory/invaded cells were removed with a cotton swab wetted in PBS, and migratory or invaded cells were fixed in 4% formaldehyde and stained with hematoxylin at room temperature. The number of migratory cells was calculated by counting cells from five fields of view per slide with 40× magnification while using a counting grid.

### 4.8. Flow Cytometry

A single-cell suspension containing 1 × 10^6^ trypsinized cells and spheres was resuspended in 1 mL of phosphate-buffered saline (PBS) and stained with fluorescent conjugated antibodies against CD133 and CD44 for 30 min. After labeling, the cells were washed three times with PBS and subsequently stained with a fluorescein isothiocyanate (FITC)- or PE-labeled secondary antibody for 30 min in the dark. The cells were analyzed using a flow cytometer (FACSCalibur; Epics Elite; Coulter Electronics, Mijdrecht, The Netherlands). Data analysis was performed using Kaluza C analysis software (https://www.beckman.com/flow-cytometry/software/kaluza, accessed on 13 August 2024) (Beckman Coulter Nederland BV, Woerden, The Netherlands). The antibodies and primers utilized are documented in Appendix A in the Appendix A.

### 4.9. Immunohistochemistry

Three subclones for IHC studies of E-cadherin, Vimentin, and N-cadherin stain. Staining intensity was classified as absent (0), mild (1+), moderate (2+), or strong (3+). The antibodies used in this study are listed in Appendix A. Cell aggregates were embedded in paraffin, sectioned (~4 μm), deparaffinized, and rehydrated. After antigen retrieval (citrate buffer, pH 6.0), sections were incubated with primary antibodies (e.g., anti-E-cadherin, anti-Vimentin, anti-N-cadherin) at 4 °C overnight, followed by HRP-linked secondary antibodies and DAB detection. Hematoxylin was used for counterstaining before microscopic examination.

### 4.10. Statistical Analysis

The independent Student’s *t*-test or ANOVA was employed to compare continuous variables between groups, while the Χ2 test was utilized for dichotomous variables. The level of statistical significance was set at *p* < 0.05. All statistical analyses were performed using SPSS version 22 software (SPSS Inc., Chicago, IL, USA).

## Figures and Tables

**Figure 1 ijms-26-01766-f001:**
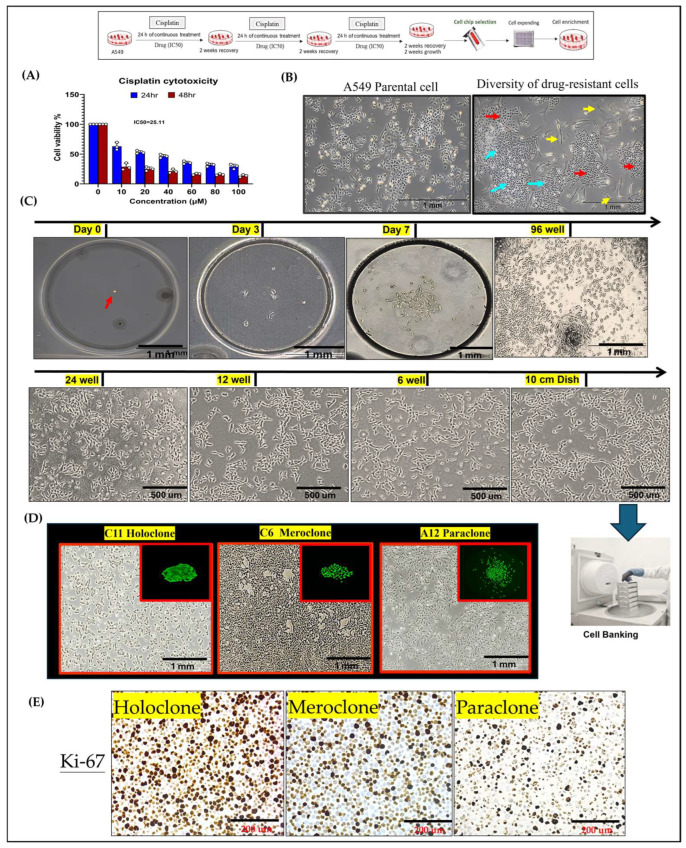
**Establishment of Phenotypically Heterogeneous Monoclonal Cell Lines via Single-Cell Cultivation.** (**A**) A549 lung cancer cells were treated with three consecutive doses of cisplatin at the IC50 concentration to simulate clinical chemotherapy pressure and induce drug resistance. The results are means ± SD for each group of cells from three separate experiments. (**B**) Following treatment, the cell population could be stratified into three phenotypically distinct subclones: tiny irregular shapes (blue arrow), polygonal shape (red arrow), and spindle (yellow arrow), reflecting pronounced intratumoral heterogeneity. (**C**) Single-cell isolation was performed using a specialized microfluidic cultivation chip. After 10–14 days, individual cell colonies reached approximately 70–80% confluence and were subsequently expanded to about 1 × 10^7^ cells. A portion of the cells was cryopreserved in liquid nitrogen for long-term storage and future analyses. (**D**) holoclones form tightly packed colonies, indicating their strong self-renewal ability; meroclones form moderately packed colonies; and paraclones form loosely packed colonies. (**E**) Ki-67 immunostaining further confirms these differences in proliferative capacity: holoclones exhibit high Ki-67 expression, meroclones display moderate levels, and paraclones show low Ki-67 expression.

**Figure 2 ijms-26-01766-f002:**
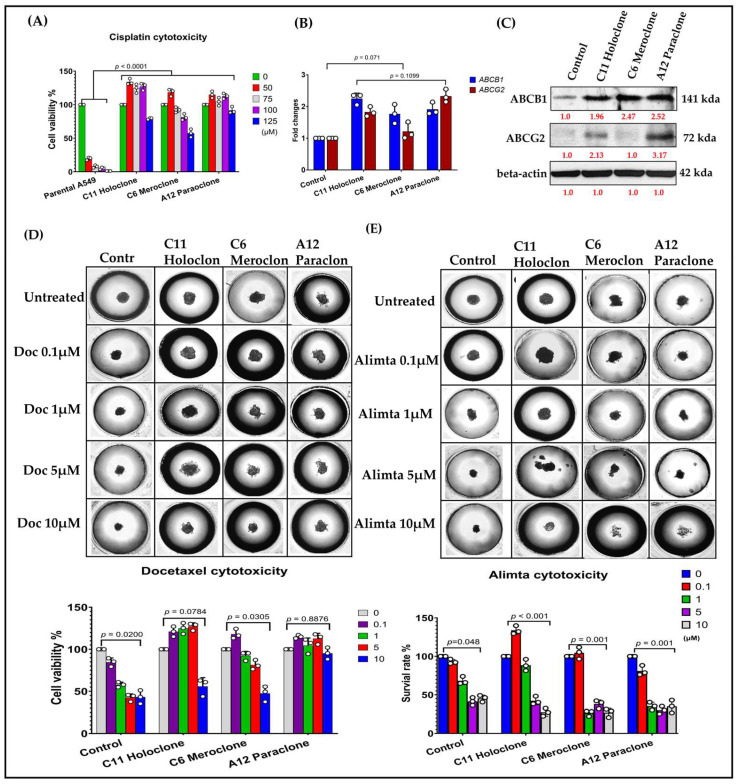
**Differential ABCB1/ABCG2 Expression and Chemosensitivity Among Drug-Resistant Subclones.** (**A**) Cisplatin sensitivity in Holoclone, Meroclone, and Paraclone clones versus parental A549 cells. (**B**,**C**) Holoclone and Paraclone cells show significantly higher *ABCB1/ABCG2* mRNA and protein levels compared to Meroclone and Control cells, indicating enhanced multidrug resistance. (**D**) In 3D spheroids, Control cells reach Docetaxel IC50 at 5–10 µM, and Meroclone achieves IC50 at 10 µM. By contrast, Holoclone and Paraclone remain insensitive within this concentration range, demonstrating greater Docetaxel tolerance. (**E**) Under Alimta (5–10 µM), all subclones achieve IC50, maintaining strong inhibitory effects. The results are means ± SD for each group of cells from three separate experiments.

**Figure 3 ijms-26-01766-f003:**
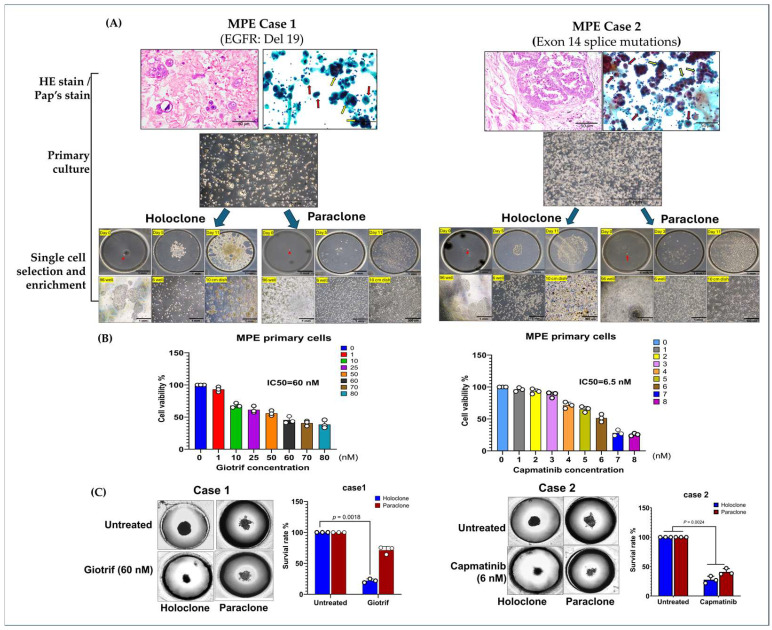
Phenotypic Heterogeneity and Tumorsphere-Based Drug Response Evidence in Primary MPE Cultures. (**A**) Hematoxylin–eosin (HE) and Papanicolaou’s staining of primary cultures revealed both clustered (yellow arrow) and individually (red arrow) dispersed cell morphologies. In primary cultures, cells displayed both elongated and floating spherical phenotypes. By applying single-cell isolation and expansion techniques, we successfully established two distinct subclones—Holoclone and Paraclone—thereby underscoring their remarkable phenotypic plasticity. (**B**) In the 3D culture system, MPE primary cells formed stable spheroids within 96 h. Drug sensitivity assays using the 3D tumorsphere model showed that Giotrif (afatinib) at 60 nM and Capmatinib at 6.5 nM achieved IC_50_. (**C**) Between the two subclones, the holoclone demonstrated marked sensitivity to Giotrif, whereas the paraclone exhibited only a limited response. In contrast, Capmatinib inhibited Holoclone and Paraclone growth, demonstrating its pronounced inhibitory efficacy against both subclone types. The results are means ± SD for each group of cells from three separate experiments.

**Figure 4 ijms-26-01766-f004:**
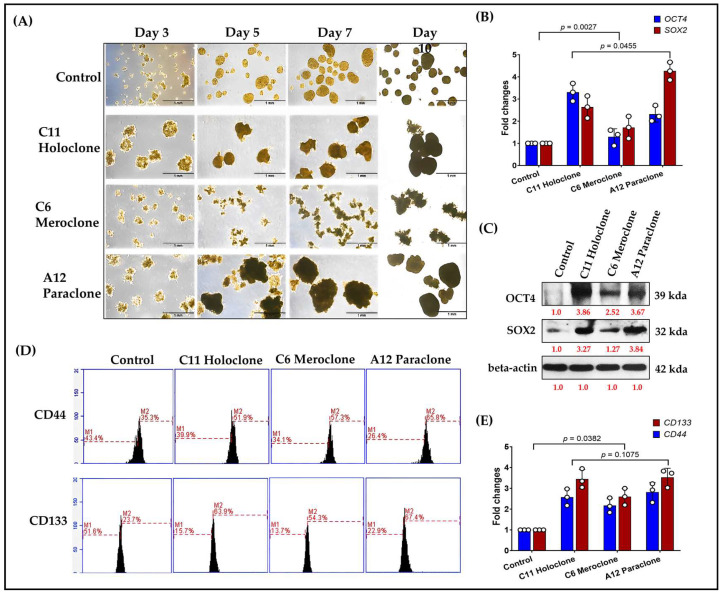
**Differential Stem Cell-Related Gene and Marker Expression in Drug-Resistant Subclones.** (**A**) After 10 days, Holoclone and Paraclone formed significantly larger spheroids than Control and Meroclone, indicating enhanced self-renewal and proliferation. Meroclone’s spiky morphology suggested an intermediate state. (**B**) RT-qPCR analyses revealed elevated *OCT4* and *SOX2* expressions in Holoclone and Paraclone, suggesting enhanced stem-like characteristics. (**C**) Western blot analyses demonstrated increased OCT4 and SOX2 protein levels in Holoclone and Paraclone, further indicating stronger stem-like traits. (**D**) Flow cytometry analysis of CD44 and CD133 expression. Representative histograms or dot plots illustrate that more than 50% of cells in the Holoclone, Meroclone, and Paraclone subpopulations are positive for CD44 and CD133, in contrast to fewer than 40% positivity in the Control group. The experiments were performed in at least three independent replicates to ensure reproducibility and statistical robustness. (**E**) RT-qPCR validation of CSC marker upregulation. Quantification of relative *CD44* and *CD133* mRNA levels in Holoclone, Meroclone, and Paraclone subclones compared with the Control group, confirming the elevated expression of these CSC-associated genes in all three drug-resistant populations. The results are means ± SD for each group of cells from three separate experiments.

**Figure 5 ijms-26-01766-f005:**
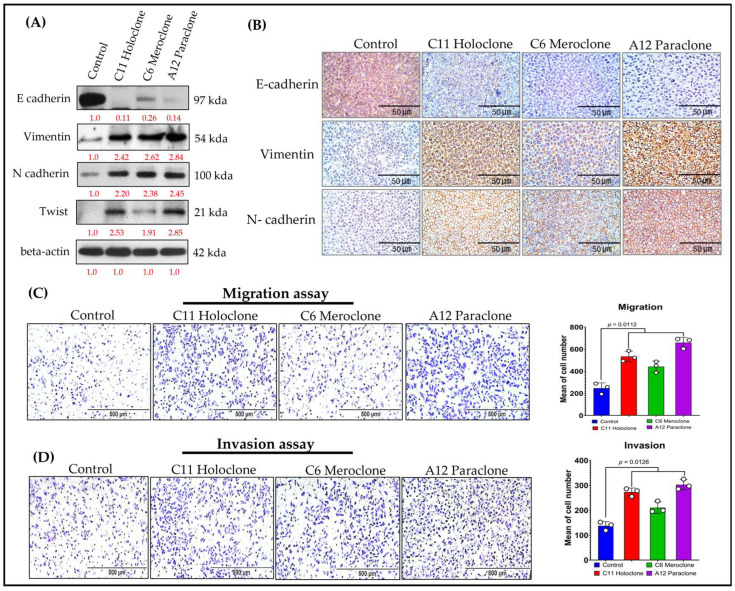
**Molecular Characterization of EMT-Driven Invasiveness in Drug-Resistant Subclones.** (**A**,**B**) Western blot and immunohistochemical analyses of Holoclone and Paraclone subclones reveal significantly higher Vimentin and N-cadherin expression levels compared to Control and Meroclone. Moreover, all three subclones (Holoclone, Meroclone, and Paraclone) exhibit increased Vimentin and N-cadherin, accompanied by reduced E-cadherin expression. (**C**) Migration assays indicate that these subclones are more migratory than Control. (**D**) Invasion assays confirm that Holoclone and Paraclone possess greater invasive potential, reflecting an EMT-driven transition. These results suggest that drug-resistant subclones adopt mesenchymal traits and Twist upregulation, enabling them to evade therapeutic pressures, enhance resistance, and potentially contribute to tumor progression, facilitating their survival and dissemination. The results are means ± SD for each group of cells from three separate experiments.

## Data Availability

The data used in this study are available upon request from the corresponding author.

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
