# Peer review of "Application of an Integrated Single-Cell and Three-Dimensional Spheroid Culture Platform for Investigating Drug Resistance Heterogeneity and Epithelial–Mesenchymal Transition (EMT) in Lung Cancer Subclones"

_ijms, 2025, doi:10.3390/ijms26041766_

Round 1
Reviewer 1 Report
Comments and Suggestions for Authors
Review of the paper titled "Application of an integrated single-cell and three-dimensional spheroid culture platform in investigating drug resistance heterogeneity and EMT in lung cancer subclones."
The authors used three-dimensional (3D) spheroid culture in conjunction with single-cell isolation techniques to study drug resistance in lung cancer. Using single-cell isolation and 3D spheroid cultures, the authors created and isolated three subclones of cisplatin-resistant A549 lung cancer cell lines (Holoclone, Meroclone, and Paraclone) to investigate the drug response patterns, molecular feature heterogeneity, and drug resistance of lung cancer. They investigated the effects of treatments on subclones using A549 cells and two MPE samples. In order to support personalized medicine strategies, the authors attempted to show how useful 3D models are for forecasting drug responses. While the 3D model is better than 2D, it's still a simplified system. The integration of single-cell and 3D cultures is a strong point, providing a more physiologically relevant model. The use of MPE samples adds clinical relevance. However, some areas need critique. The authors should discuss how factors like immune cells or stroma are missing in their model (to add as a limitation of their study).
Overall, the paper is very interesting, well designed and well written it would benefit from refinements to enhance its clarity and impact
First, I suggest to slightly modify the title as follows: "Application of an Integrated Single-Cell and Three-Dimensional Spheroid Culture Platform for Investigating Drug Resistance Heterogeneity and Epithelial-Mesenchymal Transition (EMT) in Lung Cancer Subclones.”
Methods are almost adequately described but need more details, especially for the single-cell isolation process and 3D culture protocols. Statistical analysis seems appropriate with triplicates and ANOVA, but the figure legends don't always specify the number of replicates, which is a minor issue. The protocol of q-PCR needs to be more detailed, including the method used by the authors for gene expression quantification and reference genes. In the table Table S3, add the melting temperature for each couple of primers and the size of the amplicon.
Authors are also requested to add the melting curve for each gene.
The figures show Western blots and q-PCR, so that's good, but how consistent were these results across replicates? No mention of replicates in methods, and in the figures, there are no error bars for all data points, which could be a concern.
Fig. 1B; the blue arrowhead shows three possible positions. Please change the arrow form
Line 179: “Giotrif at 60 M” → "60 µM
Line 374-375: "underscores the inadequacy of focusing solely on single factors" → "demonstrates the limitations of focusing solely on individual factors”
Various instances of inconsistent hyphenation in compound terms should be standardized (e.g., "drug-resistant" vs "drug resistant")
Figure 5B: the staining is not visible
L 172 (Fig 5 legend) “significantly higher OCT4 and SOX2”??: You mean “E-cadherin, Vimentin and N-cadherin”??
L430: “The cells were exposed to 25.11 µM cisplatin—exceeding the established IC50 concentration.". To my knowledge, the IC50 of cisplatin on A549 is less than 10 µM. Which is your reference for this value of 25 µM ??
L453: “4.3.3. D Tumorsphere Culture” à “4.3 3D Tumorsphere Culture”
L522-523: “The bodies” à “The antibodies”
Reviewer 2 Report
Comments and Suggestions for Authors
The manuscript describes the sensitivity of different subpopulations of cancer cells to chemotherapeutics. The authors used established lung adenocarcinoma A549 cells and cells established from pleural effusion of two patients with poorly differentiated pulmonary adenocarcinoma and pleural metastases. They integrated single-cell isolation and three-dimensional spheroid culture. They found clonal heteregenity in cell response to drug treatment and revealed interplay between clonal heterogeneity, CSC/EMT phenotype, and drug resistance.
I have some comments.
11. Fig. 1C, 3A: microscopic images are low quality. It's hard to see anything on them. Please, paste images in better resolution.
22. Could you introduce information about Holoclone, Meroclone, and Paraclone in the main body text of the manuscript? Have you made the characterization of the markers of proliferation or differentiation of Holoclone, Meroclone, and Paraclone to prove their proliferation/differentiation state?
33. Fig. 2A, 2D, 3B, 3C, 4B, 4E: there is no statystical analysis, and SD deviation marked on graphs. How many experimental repeats have been done in each experiment? Please provide information and correct graphs.
44. Fig.4D: the graph obtained from cytometric analysis is of low quality. It is hard to see the numbers in this figure. It would be beneficial to add an Exel chart with quantitative data of cytometric analysis.
55. Please, correct scale bar in the Fig. 1C, 3A, 4A, 5B, 5C, 5D. It is barely visible.
66. The name of the transcript should be written in italics.
77. In the Materials and Methods, section 4.8, there is a mention of assay using antibodies against CD10 and GPR77. There are no results in the text using these two antibodies. If they were not used, please delete them from the Methods.
Reviewer 3 Report
Comments and Suggestions for Authors
The manuscript titled "Application of an Integrated Single-Cell and Three-Dimensional Spheroid Culture Platform in Investigating Drug Resistance Heterogeneity and EMT in Lung Cancer Subclones" by Shin-Hu Chen et al. presents a promising approach by integrating single-cell isolation with 3D spheroid culture to explore subclonal heterogeneity and drug responses. This method aims to advance precision medicine by elucidating the relationship between heterogeneity, CSC/EMT phenotypes, and drug resistance, potentially offering a valuable tool for predicting therapeutic outcomes and guiding personalized, combination-based lung cancer treatments. However, there are key concerns that need to be addressed before the manuscript can be considered for further review.
Major:
1. Anticancer Reagents:
The manuscript mentions the resistant case of sotorasib in the introduction, however, in the experimental design, cisplatin was actually used. The authors should revise the manuscript, particularly the introduction, to reflect the use of cisplatin for resistant case instead of sotorasib.
The rationale behind the use of Docetaxel and Alimta in Section 2.2, followed by a switch to Giotrif (afatinib) and Capmatinib, should be explained more clearly. It is unclear why these changes in anticancer drugs were made, and the manuscript would benefit from providing a cohesive justification for these selections.
2. Resistant Cell Models:
The authors should reference the specific protocol they followed for establishing the cisplatin-resistant cell line.
The authors discuss Holoclone, Meroclone, and Paraclone subtypes; however, there is no clear explanation of how these subtypes were defined. The manuscript should include any known markers that are characteristic of each subtype, providing clearer criteria for their classification.
The authors should test the response of cisplatin on the Holoclone, Meroclone, and Paraclone clones and compare the sensitivity with parental A549 cells.
Additionally, it would be beneficial to have at least two clones for each cell type (Holoclone, Meroclone, and Paraclone) instead of one, even though the authors claim to have generated 17 stable cell lines. These should be reflected in the main figures to provide stronger data support.
3. Experimental Design:
The authors should consider running RNA-SEQ analysis on the Holoclone, Meroclone, and Paraclone subclones and compare their gene expression profiles with those of the parental A549 cells. Drawing conclusions about CSC/EMT phenotypes and drug resistance based solely on selected protein markers is insufficient.
Line 169 mentions that culturing cells in medium supplemented with 50% MPE yielded results, but it is unclear why this specific percentage was chosen. The authors should explain why other percentages were not tested and if 50% is the optimal condition.
The results are drawn from experiments using only one cell line. The authors should consider repeating some parts of their experiments with an additional lung cancer cell line to generalize their conclusions.
Minor:
1. Line 96: The word “unprecedented” should be removed, as single-cell selection is a commonly used laboratory technique.
2. Figure 1A: Replace the current graph with a bar chart that includes individual data points and error bars.
3. Figure 2A: Replace the current graph with a bar chart that includes individual data points and error bars.
4. Figures 2C and 2D: Provide enlarged images for each spheroid. Replace the current graphs with bar charts that include individual data points and error bars.
5. Figure 3B: Replace the current graph with a bar chart that includes individual data points and error bars.
6. Figure 3C: Provide enlarged images for each spheroid. Replace the current graphs with bar charts that include individual data points and error bars.
7. Missing Figure 4: The manuscript does not include Figure 4. Please add this figure.
8. Figures 5C and 5D: Replace the current graphs with bar charts that include individual data points and error bars.
Comments on the Quality of English LanguageShould be improved
Round 2
Reviewer 2 Report
Comments and Suggestions for Authors
Comments:
- 1, lane 123-127: need corrections, as contains descriptive information more suitable for Result part. It is better to write the title of Fig.1 and other Figures in Bold.
- Official gene symbol of MDR-1 is ABCB1. Please, correct it throughout the text.
- Lane 252, 262: OCT and SOX2 are written as gene/transcript in Italic, in general naming or protein is written by general font.
- 4C shows results of Western blot analysis of OCT and SOX2 level. There is incorrect information in the figure description, please correct.
- The description Fig.4D, 4E contain element of results, not the description of graphs. Please, edit the text. How many repeats of cytometry analyses of CD133 and CD44 markers have been performer?
- Please, make scale bar in Fig.5B more visible.
- In vivo and in vitro should be written in Italic.
- RT-qPCR, not qRT-PCR, please correct.
- [cDNA synthesis kit, e.g., High-Capacity cDNA Reverse Transcription Kit 627 (Applied Biosystems)] – square brackets are unnecessary.
- Each reaction (10 μL 630 total volume) contained [amount] μL cDNA, [final concentration] of gene-specific pri-631 mers, 5 μL of [qPCR master mix, e.g., PowerUp SYBR Green Master Mix (Thermo Fisher)], 632 and nuclease-free water. This part of text is not understable. Please correct to make text understandable to the reader.
- 1. Cell Culture, Reagent, Drug-Resistant Cancer Cell Lines and MPE Sample Prepare.
This section requires edition. It should describe methodology. Rationale for drugs using is more suitable for Introduction or Result part.
Reviewer 3 Report
Comments and Suggestions for Authors
The authors have thoroughly addressed my comments, and the manuscript is now suitable for acceptance.
Comments on the Quality of English LanguageSlight improvements in English are encouraged.
